# Culinary Medicine eConsults Pair Nutrition and Medicine: A Feasibility Pilot

**DOI:** 10.3390/nu15122816

**Published:** 2023-06-20

**Authors:** Jaclyn L. Albin, Milette Siler, Heather Kitzman

**Affiliations:** 1University of Texas Southwestern Medical Center, School of Medicine, Departments of Internal Medicine and Pediatrics, 5323 Harry Hines Blvd, Dallas, TX 75390, USA; 2Moncrief Cancer Institute and the University of Texas Southwestern Medical Center, 400 W Magnolia Ave, Fort Worth, TX 76104, USA; 3University of Texas Southwestern Medical Center, Peter O’Donnell Jr. School of Public Health, Dallas, TX 75390, USA

**Keywords:** nutrition, telemedicine, eConsult, electronic health record, interprofessional, culinary medicine

## Abstract

The global impact of diet-sensitive disease demands innovative nutrition education for health professionals and widespread, reimbursable clinical models to apply nutrition to practice. Interprofessional collaboration across disciplines and the optimization of emerging telemedicine consultation strategies, including electronic consultation (eConsult), merge to deliver vital innovation in the delivery of nutrition-based clinical care. Aligning with an existing eConsult infrastructure in the institutional electronic health record (EHR), a physician–dietitian team developed a novel Culinary Medicine eConsult. During a pilot phase, the service was introduced to primary care clinicians, and a response algorithm for eConsults was created. During the 12-month pilot phase, the Culinary Medicine team completed 25 eConsults from 11 unique primary care clinicians with a 76% (19/25) insurance reimbursement rate. Topics varied from dietary strategies for preventing and managing common metabolic diseases to specific dietary influences on microbiome health and disease flares. Requesting clinicians reported time saved in their clinic encounters and high patient satisfaction with expert nutrition guidance. EConsults in Culinary Medicine promote the integration of interprofessional nutrition care into existing clinical structures and empower enhanced access to the vital domain of dietary health. EConsults deliver timely answers to clinical questions and create opportunities for further innovation in care delivery as communities, health systems, and payors seek solutions to the growing burden of diet-sensitive diseases.

## 1. Introduction

The global impact [1,2] of diet-sensitive disease demands innovative nutrition education for health professionals and widespread, reimbursable clinical models to apply nutrition to practice. Over the past decade, strategies linking practical dietary information to clinical care have spread rapidly through the concept of Culinary Medicine [3,4,5,6,7,8,9]. The clinical application of Culinary Medicine offers innovative strategies to deliver practical, team-based care for dietary and lifestyle change promotion, ranging from the integration of recommended recipes and culinary teaching in standard patient care encounters to group cooking classes billed as shared medical appointments.

Historically, the literature focuses on Culinary Medicine in an educational context, but this term has seen an evolution and expansion of scope in recent years. Multi-disciplinary, evidence-based, and clinically relevant, the Culinary Medicine hands-on teaching kitchen model seeks to equip trainees to engage patients in personalized dietary behavior change [10,11,12,13,14]. Many medical schools across the US now incorporate Culinary Medicine, and the evidence base of impact upon community-engaged service learning and future patient care also continues to mount [15,16,17]. Advanced certification in Culinary Medicine (Certified Culinary Medicine Specialist—CCMS) [18] brings interprofessional standardization to the field, empowering its growth as a subspecialty. Additionally, the American College of Lifestyle Medicine equips clinicians to practice the pillars of lifestyle medicine through board certification (via the American Board of Lifestyle Medicine), and this training includes exposure to Culinary Medicine [19] and evidence-based medical nutrition therapy [20,21].

Many clinicians incorporate nutritional counseling and behavior change coaching into routine clinical encounters, but a core limitation is time, an oft-cited challenge to meaningful preventive healthcare [22]. Perhaps even more challenging, the vast majority of medical school graduates receive inadequate nutrition education [23,24] despite many calls for expansion [25,26,27] and are thus ill-equipped to provide clinically and culturally relevant guidance for patients navigating overwhelming and often inaccurate information about diet across traditional and social media platforms. Access to registered dietitians remains profoundly limited by poor payer reimbursement, suboptimal referral patterns, and lack of interprofessional optimization [28,29,30]. Amidst an infrastructure that hampers patient care, there is an opportunity and urgent need for innovative approaches in delivery methods for nutrition education that leads to sustainable clinical care integration.

Over the past several years, electronic consultations (eConsults) emerged as a strategy for improving access to subspecialty care. Defined as “asynchronous, consultative, provider-to-provider communications within a shared electronic health record (EHR) or web-based platform” and approved by the Centers for Medicare and Medicaid Services (CMS) as under the umbrella of telehealth benefits, eConsult implementation demonstrates increased access to medical specialists and has also been used to improve mental health access [31]. Benefits include resolving some specialty consults electronically and improving the success of scheduling in-person visits and reducing wait times, key factors particularly poignant in safety net systems [32]. Successful models of enhancing reimbursable care through eConsults create an opportunity for clinical integration of interprofessional nutrition practice.

With the growth of additional training and certification in Culinary Medicine [18,21] and the need for reimbursable models of clinical nutrition practice, eConsults provide an opportunity for disruptive innovation as physician–dietitian teams can provide interprofessional nutrition consultation. Through the unique Culinary Medicine model that focuses guidance on culturally relevant food recommendations, provision of recipes, and education about culinary techniques, the application to eConsults offers a simple, written consult report tailored to each patient’s situation. Aligning with existing models in many health systems, the implementation process requires few resources, remains a low risk, and applies to many patients across ages and conditions. The core requirements to establish this model within a health system are clinician expertise in Culinary Medicine for the consulting team and a compatible EHR structure.

## 2. Methods

To establish an eConsult process for Culinary Medicine, the physician–dietitian Culinary Medicine team initially engaged the institutional billing team as well as administrative and physician leaders to identify current systems, potential barriers, and opportunities. Due to an existing eConsult workflow across many specialties on our campus, a Culinary-Medicine-specific eConsult was built within the EHR system, creating a straightforward process for consultation that paired an open-ended consult question text box with optional EHR buttons to highlight topics where Culinary Medicine can add value. Figure 1 shows the requesting clinician’s view of the consult within one common EHR.

Next, the Culinary Medicine team identified a small pilot group of primary care clinicians and presented service details, including the process and case examples of consult topics for which the team could provide written, tailored patient care guidance in Culinary Medicine. Rooted in an informal needs assessment, the team designed a five-part response algorithm to guide each consult: (1) patient context, health background, and goals; (2) brief narrative summary, including key references for educational benefit of requesting clinician; (3) key dietary recommendations tailored to patient context and goals; (4) culinary strategies and recipes suggestions (with web links); and (5) recommendations for local resources to promote nourishing food access. After orienting the pilot primary care clinicians to the newly established eConsult workflow and clinical focus of Culinary Medicine support, the eConsult went live.

All requesting primary care clinicians completed the brief referral tailored to their patient’s area of concern, which was then electronically transmitted to the Culinary Medicine consult team. The response to the requesting clinician’s consult also occurs via the EHR, including a permanent record of the Culinary Medicine team’s consult response note in the patient’s chart and the associated billing. Consistent with the established eConsult approach defined by CMS, the requesting primary care clinician communicates the Culinary Medicine specialist’s consult advice back to the patient in either written or verbal form. Table 1 outlines the billing and coding requirements of eConsults.

## 3. Results

During a 12-month pilot phase from 1 August 2021 through 31 July 2022 of the Culinary Medicine eConsult program, the Culinary Medicine physician–dietitian team completed 25 eConsults. The team received at least one consult per month and a maximum of four consults in a single month. Many consult topics overlapped between patients, enabling the consulting team to repurpose prior content while individualizing recommendations in the context of each patient’s overall health circumstances.

The consult team developed a database of key information, resources, and recommendations and stored topic-specific consults on an institutionally approved cloud storage site without protected health information. Common consult themes led to efficient responses tailored to the individual needs of each patient while also delivering a return on prior investment of effort and time. Common recurring consultation topics included dietary strategies to support the treatment of metabolic and other chronic conditions, such as diabetes, fatty liver disease, and irritable bowel syndrome, as well as guidance for general health promotion, including reducing chronic inflammation and maintaining a healthy weight. Unique and more complex consultation topics comprised the influence of diet and microbiome health on specific conditions, such as eczema and rosacea, as well as specific dietary needs for patients with physical disabilities or severe dietary allergies.

During the pilot year, a total of 11 unique primary care clinicians initiated the 25 eConsults, with 4 of those clinicians requesting consults for several different patients. The assessment of billing data showed that 76% (19 of 25) of consults were reimbursed by payors, including Medicare and various private insurance companies. Documentation of patient consent by the original requesting clinician is a requirement for the consult to be billable, and rejected claims lacked this necessary documentation of consent.

Qualitative feedback from requesting primary care clinicians consistently demonstrated the importance of this unique clinical support resource. Requesting clinicians cited the benefit of an eConsult when running out of time during a clinic visit and the convenience of the straightforward process of copying and pasting the response content into a patient portal message. Other key feedback included the ability to highlight the unique training and expertise of the Certified Culinary Medicine Specialist team, noting that some patients take nutrition and culinary guidance more seriously than those with specific additional credentialing and expertise.

## 4. Discussion

EConsults in Culinary Medicine offer the opportunity to brand Culinary Medicine as clinical care, even in the absence of resources for broader integration, such as individual Culinary Medicine consult appointments or shared medical visits. Expert consultation for nutrition concerns—from the routine to the complex—validates the vital role that diet plays in health promotion and the expertise necessary to provide evidence-based guidance. Additionally, the Culinary Medicine eConsult is collaborative at its foundation, with a cornerstone partnership between a physician and a registered dietitian nutritionist (RDN) as a core feature. This multi-disciplinary approach promotes a well-rounded, interprofessional perspective, and opportunities abound for collaboration in educational, research, and clinical domains. Clinician billing in nutritional domains can demonstrate the successful augmentation of reimbursement for this vital work and open doors to advocate for consistent reimbursement across professions, particularly in the field of dietetics.

As a result of successful eConsult implementation, which demonstrated both primary care clinician and patient demand for this clinical support, the Culinary Medicine physician-dietitian team at our institution was asked to expand and develop a full Culinary Medicine Clinical Service Line. The eConsult model thus served as a gateway to additional services, including in-person Culinary Medicine consultations and shared medical visit cooking classes. The establishment of Culinary Medicine as an interprofessional clinical service that is billable and reimbursable was a vital first step to gaining stakeholder support for growth. Further, although medical nutrition therapy (MNT) is an effective intervention, it is time-intensive, and many eligible patients do not utilize this service [33]. Physicians also may not refer to MNT frequently due to time barriers preventing explanation of the importance of nutrition to disease management and prevention as well as describing MNT to patients [29,30]. Thus, the eConsult is a practical and time-efficient mechanism to encourage patients and physicians to integrate nutrition strategies to improve health outcomes and quality of life.

### 4.1. Limitations

The application of the eConsult model necessitates institutional awareness and the implementation of the process of electronic consultation as well as billing negotiation and structure. Each institution varies, but the authors recognize this as a limitation to scalability. In the context of an existing eConsult structure, the primary limitation for application to Culinary Medicine is the necessity of an appropriately trained team to serve as the consultants.

It is important to note that the Culinary Medicine eConsult is in no way intended to be a replacement for nutrition assessment and medical nutrition therapy conducted by a registered dietitian. Although most e-Consults recommend additional one-on-one sessions with a dietitian, current barriers to access can make navigation to a dietitian beyond the e-Consult difficult, and most patients in our system do not receive any nutrition-related care. As such, the culinary focus of the eConsult program provides practical, accessible support strategies to patients and primary care clinicians while also advocating for additional nutrition support when indicated.

An additional limitation worth mentioning is the small sample size of our pilot feasibility study. This brief report introduces a novel concept pairing the relatively new process of eConsults for improved access to specialist input with the emerging clinical practice of Culinary Medicine. As such, we acknowledge that 20 patient consults and 11 requesting primary care clinicians represent a small volume of initial participants, and further work on the application and scalability of eConsults is needed.

### 4.2. Future Directions

EConsults in Culinary Medicine provide an accessible, broadly relevant strategy for clinical teams with Culinary Medicine expertise to provide evidence-based guidance to primary care clinicians and their patients struggling to get access to nutritional support. This guidance serves to educate and guide the requesting clinician and ideally delivers tailored and actionable culinary nutrition guidance for the patient in a format that can be easily communicated via the electronic health record (through patient portal messaging, a printed letter or handout, or similar modalities). This approach utilizes an existing, insurance-covered clinical workflow strategy to build a foundation of Culinary Medicine therapy as an interprofessional clinical service line.

For many patients, eConsults can serve as their first interaction with a registered dietitian and can operate as a gateway, highlighting the potential benefits of additional nutrition clinical services. As the current literature suggests, some eConsults definitively answer clinical questions requiring no further interactions. In other cases, they provide a bridge to additional intervention, ensuring that the appropriate team engages in a patient’s ongoing care. For patients with nuanced nutritional needs, such as guidance for a new food allergy diagnosis or celiac disease, an eConsult may lead to a recommendation for such dietitian-led, individualized nutritional guidance. Other patients may benefit from referral to specific community resources or programs to support consistent access to nourishing food, including local food pantry programs, food prescription programs, or federal nutrition programs. Complex consultations also reveal opportunities for further engagement, particularly personalized medical nutrition therapy by referral to a registered dietitian nutritionist, one-to-one consults with the Culinary Medicine team, or group cooking classes via billable shared medical appointments. This growing model of care can take place in a teaching kitchen where a patient learns culinary skills and builds self-efficacy for dietary change, a method with the potential to revolutionize the prevention and treatment of chronic disease.

## 5. Conclusions

Applying the established approach of eConsults, this intervention sought to increase access to culinary and nutritional support for patients while building Culinary Medicine as an interprofessional and reimbursable clinical care strategy. During this pilot study, the development of a Culinary Medicine eConsult service led by a physician–dietitian team was feasible both in process and reimbursement. Positive feedback from the participating primary care clinicians who requested Culinary Medicine eConsults showed interest and engagement in offering culinary nutrition to support their patients.

Further, the Culinary Medicine eConsult is a promising tool that highlights the beneficial synergy of physician–dietitian partnerships in the food as medicine area of practice. Although the eConsult is not designed to replace traditional one-on-one clinical visits with either a physician or a dietitian, it can make a powerful auxiliary tool. In a current climate of overly taxed clinicians as well as scheduling difficulties and staffing shortages, innovative tools that help connect patients with resources and care in an efficient, timely, and reimbursable manner are sorely needed.

To achieve successful progress in mitigating the tsunami of diet-related disease, an all-hands-on-deck strategy that emphasizes interprofessional teamwork is vital. Physician education in nutrition and subsequent engagement in clinical integration enhances reimbursement potential, elevating the vast importance of this field. As the evidence of impact grows and becomes impossible to ignore by payor models, clinicians, and patients alike, teams should be ready to bring everyone to their collaborative table, sharing the potential of food as medicine.

## Figures and Tables

**Figure 1 nutrients-15-02816-f001:**
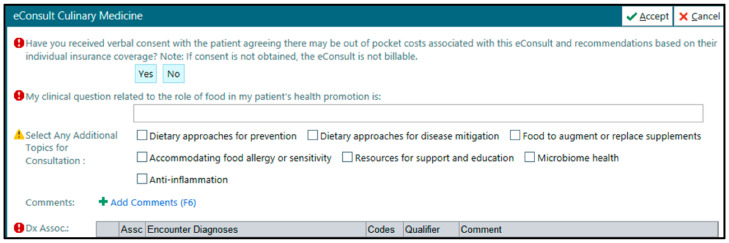
Culinary Medicine eConsult EHR referral example, requesting clinician view.

**Table 1 nutrients-15-02816-t001:** Interprofessional eConsult process and practical details.

Requesting Clinician: Consult sent via EHR for new or established patient or problem; Must document patient consent (essential for reimbursement); Time reflects preparation, consult, and subsequent communication to patient; Billing code 99452 reflects >15 min of time and represents 0.7 Relative Value Units (RVUs).
Consulting Clinician: Receives EHR-based eConsult; Cannot provide eConsultation to the same patient more than once in 7 days; Cannot see the same patient face-to-face within 14 days of eConsult; Must document eConsult in electronic written record; Billing code 99451 reflects >5 min of time and represents 0.7 RVUs.

## Data Availability

The data presented in this study are available on request from the corresponding author. The data are not publicly available due to the nature of being a feasibility pilot without funding or public database access.

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
