# Peer review of "Culinary Medicine eConsults Pair Nutrition and Medicine: A Feasibility Pilot"

_nutrients, 2023, doi:10.3390/nu15122816_

Round 1
Reviewer 1 Report
In this article Albin et al. discuss culinary medicine and put forward eConsults as a way to empower Nutrition in practice. While well presented there are a few issues that need addressing:
1. I find the use of the word culinary medicine is miss represented. What you have discussed is as referral prompt, how does that constitute medicine or treatment?
2. This reinforced on page 4 line 154-158 Limitations, that's because what you are saying is in no way medicinal or allied treatment, so no it would in no way come close to replacing dietitians.
3. It is not clear who the 11 unique clinicians are? is it the kitchen manager that is doing this culinary medicine?
4. It is extremely dangerous to create new words around the delivery of and practice of primary health care. You state in future directions page 4 line 161 that this will 'provide-evidenced-based guidance to other clinicians? How by ticking a referal box?
5. Your evidence in no way supports what you are stating.
Reviewer 2 Report
This short report is interesting and touches on an interesting, contemporary topic, but I have a few comments:
My comments:
1. I do not see a close relationship between the content of the report and its title - maybe the title should be changed
2. The analysis is carried out on a small group - this may make conclusions difficult - because it is difficult to expect that the authors increase the size of the study group, they should somehow take this into account in the analysis, discussion and conclusions
3. The discussion is very short and not really a discussion - the authors summarized their research but did not critically discuss their results with the results of other authors in the same or very similar research area. This section should be rewritten in the format necessary for discussion.
4. Conclusions are the free thoughts of the authors related to the discussed topic, but do not result directly from the presented results - in my opinion, it is necessary to formulate the conclusions again.
Round 2
Reviewer 2 Report
I find this manuscript interesting. The idea of eConsultation in this area may be a good solution in today's digitized society. I don't have specific comments, only one: in line 86 the abbreviation EMR appears, which in the main text is explained only in line 92 - the explanation should be the first time the abbreviation is used (does not apply to the abstract) - please correct it.